# Decitabine-Mediated Upregulation of CSPG4 in Ovarian Carcinoma Cells Enables Targeting by CSPG4-Specific CAR-T Cells

**DOI:** 10.3390/cancers14205033

**Published:** 2022-10-14

**Authors:** Dennis Christoph Harrer, Charlotte Schenkel, Carola Berking, Wolfgang Herr, Hinrich Abken, Jan Dörrie, Niels Schaft

**Affiliations:** 1Department of Hematology and Internal Oncology, University Hospital Regensburg, 93053 Regensburg, Germany; 2Department of Dermatology, Friedrich-Alexander-Universität Erlangen-Nürnberg, Universitätsklinikum Erlangen, Hartmannstraße 14, 91052 Erlangen, Germany; 3Leibniz Institute for Immunotherapy, Division of Genetic Immunotherapy, University Regensburg, 93053 Regensburg, Germany; 4Comprehensive Cancer Center Erlangen European Metropolitan Area of Nuremberg (CCC ER-EMN), Östliche Stadtmauerstraße 30, 91054 Erlangen, Germany; 5Deutsches Zentrum Immuntherapie (DZI), Ulmenweg 18, 91054 Erlangen, Germany

**Keywords:** oncology, immunotherapy, antigen, SKOV-3, CAR-T cells, CSPG4, decitabine, ovarian cancer

## Abstract

**Simple Summary:**

Cancer therapy by specifically redirected T cells has revolutionized the field of oncology. However, the repertoire of targetable antigens is small. Here, we use the FDA-approved drug decitabine to upregulate the surface antigen CSPG4 on CSPG4-negative ovarian carcinoma cells. By optimizing decitabine dosing, we converted more than 50% of treated ovarian carcinoma cells to CSPG4-positive cells. Importantly, CSPG4 is a very well-established target antigen in melanoma, and we could previously demonstrate that T cells engineered to target CSPG4 could reliably kill CSPG4-positive melanoma cells. Using CSPG4-specific T cells, we demonstrate CSPG4-directed killing of decitabine-treated ovarian carcinoma cells, thereby adding CSPG4 to the repertoire of target antigens for ovarian cancer.

**Abstract:**

The addition of CAR-T cells to the armamentarium of immunotherapy revigorated the field of oncology by inducing long-lasting remissions in patients with relapsing/refractory hematological malignancies. Nevertheless, in the lion’s share of patients diagnosed with solid tumors, CAR-T-cell therapy so far failed to demonstrate satisfactory anti-tumor activity. A crucial cause of resistance against the antigen-specific attack of CAR-T cells is predicated on the primary or secondary absence of suitable target antigens. Thus, the necessity to create a broad repertoire of different target antigens is vital. We aimed to evaluate the potential of the well-established melanoma antigen chondroitin sulfate proteoglycan 4 (CSPG4) as an inducible antigen in ovarian cancer cells, using CSPG4-negative SKOV-3 ovarian cancer cells as a model. Based on the hypomethylating activity of the FDA-approved drug decitabine, we refined a protocol to upregulate CSPG4 in the majority of decitabine-treated SKOV-3 cells. CSPG4-specific CAR-T cells generated by mRNA-electroporation showed CSPG4-directed cytokine secretion and cytotoxicity towards decitabine-treated SKOV-3. Another ovarian cancer cell line (Caov-3) and the neoplastic cell line 293T behaved similar. In aggregate, we generated proof-of-concept data paving the way for the further exploration of CSPG4 as an inducible antigen for CAR-T cells in ovarian cancer.

## 1. Introduction

During the last decade, the advent of CAR-T-cell therapy has ignited enthusiasm in the field of oncology. Using T cells genetically engineered to target tumor cells by a chimeric antigen receptor (CAR), long-lasting complete remissions were achieved in patients with relapsing or refractory hematological malignancies [1]. To date, FDA-approved CAR-T-cell products exist for various lymphoma entities, multiple myeloma, and acute lymphoblastic leukemia [2]. Nevertheless, in patients with solid tumors CAR-T-cell therapy has shown limited efficacy so far [3].

The scarcity of suitable target antigens poses a crucial obstacle to the success of CAR-T cells in the realm of solid tumors [4]. Many target antigens on solid tumors are also broadly expressed by healthy cells, thereby harboring the risk of serious on-target/off-tumor toxicities [5]. Moreover, a crucial mechanism of resistance mounted by tumor cells against the antigen-specific attack by CAR-T cells is based on the loss or downregulation of target antigens [6]. In this situation, a targetable antigen selectively expressed by cancer cells in sufficiently high levels is needed.

Chondroitin sulfate proteoglycan 4 (CSPG4), formerly called melanoma-associated chondroitin sulfate proteoglycan (MCSP) or high-molecular-weight melanoma-associated antigen (HMW-MAA), has attracted attention as a target structure for CAR-T-cell therapy in recent years due to a number of reasons [7], including the expression on a broad variety of malignancies, such as melanoma, glioblastoma, leukemia, and breast cancer [7]. In addition, CSPG4 expression was found on cancer-associated vasculature, which would render the tumor microenvironment sensitive to CSPG4-directed CAR-T cells [8]. Importantly, accumulating evidence implicates a role of CSPG4 in tumor evolution by providing growth signals and by counteracting apoptotic stimuli [9]. Moreover, CSPG4 promotes invasion and metastasis formation through interactions with the extracellular matrix and tyrosine kinase signaling [10]. In sum, CSPG4 fosters crucial steps in tumor progression, thereby possibly conferring growth advantages on CSPG4-positive tumor cells. Due to its involvement in tumor progression, CSPG4 is assumed to be less sensitive to antigen-shutdown in response to targeting via CSPG4-specific CAR-T cells.

In melanoma cells, which usually show a uniform expression of CSPG4, Luo et al. demonstrated that CSPG4 expression may be impeded by intense methylation of its promotor [11]. Making use of decitabine (5-aza-2′-deoxycytidine), which is a well-characterized demethylating substance, a dose-dependent upregulation of CSPG4 on several melanoma cell lines was observed [11]. Meanwhile, decitabine has been approved by the FDA to treat myeloid malignancies. Just recently, Leik et al. harnessed the demethylating activity of azacitidine, a close structurally relative of decitabine, to increase the antigen-density of CD70 on malignant myeloid cells in order to facilitate targeting by CD70-specific CAR-T cells [12]. However, de novo upregulation of a tumor antigen on previously negative malignant cells using a demethylating agent followed by antigen-specific targeting has never been reported.

Here we use decitabine to induce expression of CSPG4 on SKOV-3 ovarian carcinoma cells. Metastasized ovarian cancer is a hard-to-treat cancer entity associated with a dismal prognosis if refractory to frontline chemotherapy [13]. Hence, an expansion of the treatment armamentarium is needed. Over the past years, CSPG4-CAR-T cells generated via mRNA-electroporation were well characterized by us and others as a therapeutic option to treat melanoma [14,15,16,17].

The goal of the current study was to employ CSPG4-CAR-T cells generated by mRNA-electroporation to target CSPG4-negative SKOV-3 ovarian carcinoma cells upon decitabine-mediated up-regulation of CSPG4. In addition, we wish to raise awareness for CSPG4 as a potentially inducible antigen in solid tumors.

## 2. Materials and Methods

### 2.1. Cells and Reagents

Peripheral blood mononuclear cells (PBMCs) were isolated from whole blood obtained from healthy donors upon informed consent and approval by the institutional review board using density centrifugation on Lymphoprep (Axis-Shield, Oslo, Norway). Following extraction, PBMCs were cryopreserved and stored at −80 °C until experimental use. T cells were cultured in RPMI 1640 + GlutaMAX (Gibco, ThermoFisher), 100 IU/mL penicillin + 100 µg/mL streptomycin (Pan-Biotech, Aidenbach, Germany), 2 mM HEPES (PAA, GE healthcare), and 10% (*v*/*v*) heat-inactivated fetal calf serum (Pan-Biotech, Aidenbach, Germany).

Target cell lines encompassed 293T cells (human embryonic kidney cells that express the SV40 large T antigen) and SKOV-3 human ovarian cancer cells (ATCC HTB77; American Type Culture Collection, Manassas, VA). Caov-3 ovarian cancer cells were a kind gift from the chair of immunology at the Leibniz-institute for immunotherapy. The cells were maintained in DMEM + GlutaMAX (Gibco, ThermoFisher), 100 IU/mL penicillin + 100 µg/mL streptomycin (Pan-Biotech, Aidenbach, Germany), and 10% (*v*/*v*) heat-inactivated fetal calf serum (Sigma-Aldrich, St. Louis, MO, USA).

### 2.2. Treatment with Decitabine

Decitabine (Sigma-Aldrich, St. Louis, MO, USA) was dissolved in phosphate-buffered saline (PBS), and aliquots were stored at −80 °C until experimental use. For all experiments, freshly thawed aliquots were used. After thawing, the remaining decitabine was discarded. For dose response and time course experiments, SKOV-3 cells were seeded at 5 × 10^5^ cells in 24 wells filled with 1 mL of supplemented DMEM medium. Decitabine was added at the indicated concentrations. PBMC and CAR T-cells were seeded at 1 × 10^6^ per ml and decitabine was added at 1 μM daily for 6 days. In case of daily decitabine addition, the medium was changed completely every day.

### 2.3. T-Cell Expansion

Peripheral blood mononuclear cells were thawed and directly activated with 0.1 µg/mL anti-CD3 antibody OKT3 (Orthoclone OKT3; Jannsen-Cilag, Neuss, Germany). The ensuing T-cell expansion was previously reported in detail [18]. Interleukin-2 (Proleukin; Novartis, Nuremberg, Germany) was supplemented on days 0, 2, 3, 5, and 7. After 10 days, T cells were subjected to mRNA-electroporation.

### 2.4. Flow Cytometry

CSPG4 expression on 293T and SKOV-3 cells was detected using an anti-human CSPG4 antibody (BD Biosciences, Franklin Lakes, NJ, USA, clone: 9.2.27). IgG2a isotype-staining served as control. For live/dead staining the eFluor780 viability dye (ThermoFisher) was used. Surface expression of the CAR was analyzed via flow cytometry 24 h after electroporation. The CAR was stained with the goat-F(ab’)2 anti-human IgG antibody (Southern Biotech, Birmingham, AL, USA) directed against the extracellular IgG1 CH2CH3 (Fc-spacer) CAR domain. Immunofluorescence was measured using a BD FACSLyric (BD Biosciences, Heidelberg, Germany) equipped with FACSuite software (BD Biosciences). Data were analyzed using FlowJo software version 10.7.1 Express 5 (BD Biosciences).

### 2.5. mRNA-Electroporation

A second-generation CAR (CSPG4_HL_-CD28/CD3ζ-CAR) directed against CSPG4 (chondroitin sulfate proteoglycan 4) was expressed in T cells. The structure of this chimeric antigen receptor was previously published in detail [17]. RNA-transfection was performed as reported elsewhere [19]. T cells were either mock-electroporated (no RNA) or transfected with 15 μg of RNA encoding the CSPG4-specific CAR employing a Gene Pulser Xcell (Bio-Rad, Hercules, CA, USA) at 500 V (square wave pulse) for 5 ms. After transfection, T cells were cultured in supplemented RPMI medium. In vitro transcription of RNA was carried out as previously described in detail [15].

### 2.6. Cytokine Secretion

Target cells comprising CSPG4-negative 293T cells and SKOV-3 cells with and without decitabine pre-treatment were seeded in 96-well round-bottom plates (25 × 10^3^ cells per well) overnight, before addition of mock-T cells or CSPG4-specific CAR-T cells (24 h after electroporation) at 1 × 10^5^ cells per well in 200μL total volume. Supernatants were harvested after 48 h of co-culture. Subsequently, IFNγ in culture supernatants was detected by ELISA using a solid-phase anti-IFNγ capture antibody (1 μg/mL) together with the biotinylated anti-IFNγ detection antibody (0.5 μg/mL) (BD Biosciences). Finally, visualization of the reaction product was performed with a peroxidase–streptavidin conjugate (1:10,000) and ABTS (Roche Diagnostics, Indianapolis, IN, USA).

### 2.7. Cytotoxicity Assay

Target cells comprising CSPG4-negative 293T cells, SKOV-3 cells with and without decitabine treatment, and Caov-3 cells with and without decitabine treatment were seeded in 96-well round-bottom plates (5 × 10^3^ cells per well) overnight before addition of mock-T cells or CSPG4-specific CAR-T cells (24 h after electroporation) at the indicated effector to target ratios. For some experiments, mock-T cells and CAR-T cells were incubated directly after electroporation with decitabine at 1 μM for 20 h and throughout subsequent co-incubation with SKOV-3 cells with and without decitabine treatment. After 48 h of co-culture, specific cytotoxicity was determined via an XTT-based colorimetric assay employing the Cell Proliferation Kit II (Roche, Mannheim, Germany). Viable tumor cells in triplicate experimental wells were counted according to the following formula: viability (%) = [OD (experimental wells—corresponding number of T cells)]/[OD(tumor cells without T cells − medium)] × 100, where OD is optical density. Cytotoxicity (%) was defined as 100—viability (%). The viability of target cells was determined as the mean values of twelve wells containing only target cells subtracted by the mean background level of wells containing medium only.

### 2.8. Statistical Analysis

Statistical analysis was executed using GraphPad Prism, Version 9 (GraphPad Software, San Diego, CA, USA); *p* values were calculated by Student’s *t*-test, * indicates *p* ≤ 0.05, ** indicates *p* ≤ 0.01, and *** indicates *p* ≤ 0.001.

## 3. Results

### 3.1. Decitabine Mediates Dose-Dependent Upregulation of CSPG4 on SKOV-3 Ovarian Cancer Cells

SKOV-3 ovarian cancer cells are broadly considered CSPG4-negative [20]. In order to verify the absence of CSPG4 on SKOV-3 cells, we utilized the 9.2.27 anti-CSPG4 antibody, which binds to a broad spectrum of isoforms with diverse glycosylation patterns. In accordance with previous reports, no CSPG4-expression was detected on SKOV-3 cells by flow cytometry (Figure 1a). Seeking to render SKOV-3 cells sensitive to CSPG4-targeting therapies, we treated SKOV-3 cells with escalating doses of decitabine. Importantly, the decitabine dosing range is predicated on mean decitabine blood levels attained with clinically applied decitabine infusion regimens [21]. After a single application of increasing doses of decitabine, a dose-dependent upregulation of CSPG4 was confirmed by flow cytometry 5 days later (Figure 1b,c). A dose of 0.5 µM induced a significant upregulation of CSPG4 as compared to the PBS control (Figure 1c). With decitabine concentrations of 6 µM or 8 µM, more than one-third of SKOV-3 cells could be converted to CSPG4-positive cells (Figure 1d).

Taken together, a single application of decitabine could induce expression of CSPG4 in SKOV-3 ovarian cancer cells implying a display of a potential target by CSPG4-directed approaches.

### 3.2. Daily Application of Decitabine Mediates Superior Upregulation of CSPG4 on SKOV-3 Ovarian Cancer Cells

We next aimed at refining decitabine-mediated CSPG4-expression to optimize the potential targetability of SKOV-3 cells with CSPG4-CAR-T cells. Decitabine, is clinically administered as daily intravenous infusion for 5 consecutive days for the treatment of myeloid malignancies [22]. We adopted this regimen and compared CSPG4-upregulation mediated by daily decitabine treatment over 5 days with decitabine administered as a single shot (Figure 2a). Flow cytometry after 5 days, once again, revealed dose-dependent CSPG4 upregulation upon single-shot decitabine stimulation (Figure 2b). Moreover, using daily application of decitabine greater CSPG4 upregulation was observed, with statistical significance reached for 0.5 μM, 1 μM, and 2 μM (Figure 2b). Contrary to single-shot application, the highest CSPG4 upregulation with daily decitabine application was mediated with 1 μM, and lower upregulation was observed with higher concentrations (Figure 2b,c). Correspondingly, daily decitabine application conferred higher percentages of CSPG4-positive cells culminating at around 50% CSPG4-positve cells in response to daily application of 1 μM decitabine (Figure 2c). Hence, this concentration was employed in all ensuing experiments.

In sum, we could refine decitabine-mediated CSPG4 upregulation by decitabine dose-titration and comparing single-shot versus daily application.

### 3.3. Kinetics of Decitabine-Mediated CSPG4-Upregulation on SKOV-3 Ovarian Cancer Cells

Next, we sought to identify the time point showing the highest CSPG4 upregulation after decitabine addition. To this end, SKOV-3 ovarian cancer cells were treated daily with 1 μM decitabine for 14 days, and CSPG4 expression was monitored daily by flow cytometry (Figure 3a). Significant CSPG4 expression was observed from the first day of decitabine treatment onwards until day 14 (Figure 3b). The highest CSPG4 expression was documented five to six days after the onset of decitabine treatment, followed by a steady but slow decline towards day nine (Figure 3b). At the point of highest CSPG4 expression on days five to eight, approximately 60% of all SKOV-3 cells upregulated CSPG4 (Figure 3c). After fourteen days of decitabine treatment, the percentage of CSPG4-positive SKOV-3 cells dropped to approximately 45% (Figure 3c,d). Control SKOV-3 cells, which were not exposed to decitabine, but were treated with PBS, did not show any tangible CSPG4 expression at any point in time (Figure 3b). Seeking to validate this dosing regimen for CSPG4 upregulation in other cell lines, we used Caov-3 ovarian cancer cells previously reported to be CSPG4-negative [23], and 293T cells. While Caov-3 cells exhibited a very low CSPG4 baseline expression, untreated 293T cells did not show CSPG4 expression at all (Appendix A). However, after treatment with decitabine at 1 μM for 6 days CSPG4 could be significantly upregulated both in Caov-3 cells (about 40% CSPG4-positive cells) (Appendix A) and in 293T cells (about 50% of CSPG4-positive cells) (Appendix A). This corroborates the basic concept of decitabine-mediated CSPG4 upregulation in neoplastic cells, with special emphasis on ovarian cancer cells with minimal CSPG4 expression.

Collectively, daily addition of decitabine for 6 days converts more than half of treated SKOV-3 cells to CSPG4-positive cells, which may render those cancer cells potentially sensitive to CSPG4-directed therapies, such as CSPG4-specific CAR-T cells.

### 3.4. Decitabine-Mediated Upregulation of CSPG4 on SKOV-3 Ovarian Cancer Cells Mediates Recognition by CSPG4-Specific CAR-T Cells

SKOV-3 cells were treated daily with 1 μM decitabine for 6 days before co-culture with CSPG4-CAR-T cells (Figure 4a). A second-generation CD28-co-stimulated CAR was used (Figure 4b), the antigen-binding domain of which was derived from the 9.2.27 antibody used to stain CSPG4 throughout this study. T cells redirected to CSPG4 were generated by mRNA-electroporation of CAR-encoding RNA, following expansion of bulk T cells using OKT3 and IL-2. Mock (electroporation without RNA) T cells served as controls. Uniform expression of the CSPG4-CAR was confirmed 24 h after electroporation by flow cytometry (Figure 4c). Thereafter, CAR-T cells and mock-T cells were incubated with 293T cells, which do not express CSPG4, (Appendix A), SKOV-3 cells treated with decitabine, and SKOV-3 cells treated with PBS. After 48 h of co-culture, IFNγ in the supernatant was quantified by via ELISA. CSPG4-CAR-T cells exhibited significant IFNγ-production in response to decitabine-treated CSPG4-positive SKOV-3 cells (Figure 4d). In contrast, PBS-treated CSPG4-negative SKOV-3 cells, and CSPG4-negative 293T cells did not induce IFNγ-production in CSPG4-CAR-T cells (Figure 4d). Mock-CAR-T cells serving as negative control did not evince IFNγ-production in response to the target cells (Figure 4d). Corresponding results were obtained regarding cytotoxicity of CSPG4-CAR-T cells towards 293T cells and SKOV-3 cells either treated with PBS or decitabine (Figure 4e). Across various effector to target ratios, CSPG4-CAR-T cells displayed significant cytotoxicity towards decitabine-treated SKOV-3 cells, while 293T cells and PBS-treated SKOV-3 cells were not killed (Figure 4e). Mock-T cells did not display any cytotoxicity towards any target cell line (Figure 4e). Finally, we assayed CSPG4 upregulation on PBMCs and CAR-T cells to estimate the risk for off-target toxicity to blood cells as well as the potential of fratricide among CSPG4-specific CAR-T cells. Neither PBMCs nor CAR-T cells showed CSPG4 upregulation in response to treatment with decitabine rendering CSPG4-directed targeting of PBMCs and CAR-T cells unlikely (Appendix A). Moreover, we confirmed that CSPG4-CAR-T cells retain their cytolytic capacity towards decitabine-treated SKOV-3 ovarian cancer cells in the presence of 1 μM decitabine enabling a simultaneous application of CAR-T cells together with decitabine in the clinical setting (Appendix A).

To generalize our findings, we examined the susceptibility of both 293T cells and Caov-3 cells for antigen-specific killing at various effector to target ratios by CSPG4-CAR-T cells after decitabine treatment (Appendix A). While both 293T cells and Caov-3 cells without decitabine treatment did not evoke cytotoxicity from CSPG4-CAR-T cells, a substantial and significant cytotoxicity was observed at all effector to target ratios, when the targets had been treated with decitabine (Appendix A). Hence, the use of decitabine to render tumor cells susceptible to killing by CSPG4-CAR-T cells is not exclusively restricted to SKOV-3 cells, but works also in other ovarian cancer cells, such as Caov-3, and neoplastic cells such as 293T.

In sum, decitabine-mediated upregulation of CSPG4 on SKOV-3 ovarian cancer cells enables antigen-specific targeting using CSPG4-CAR-T cells resulting in effective CSPG4-directed target cell killing even in the presence of decitabine.

## 4. Discussion

In the present work, we explored the potential of CSPG4 as an inducible secondary target antigen for CAR-T-cell therapy of solid tumors using SKOV-3 ovarian cancer cells as targets. First, we demonstrated dose-dependent decitabine-mediated upregulation of CSPG4 on SKOV-3 ovarian cancer cells. Next, we refined the process of decitabine-mediated CSPG4-upregulation by dose-titration and kinetics analyses to convert more than 50% of treated SKOV-3 cells to CSPG4-positive targets. Finally, bulk T cells were equipped with a CSPG4-specific CAR, and antigen-specific targetability of decitabine-treated SKOV-3 ovarian cancer cells by CSPG4-specific CAR-T cells was demonstrated. To corroborate our findings, key experiments were successfully repeated with another ovarian cancer cell line and a neoplastic cell line. This is the first study to show targeting of CSPG4 on tumor cells after decitabine-mediated upregulation raising the hypothesis for CSPG4 as an inducible secondary antigen on solid tumors with special emphasis on ovarian cancer.

Originally, decitabine-mediated upregulation was reported in melanoma cell lines more than a decade ago [11]. Seeking to investigate whether promotor methylation regulates the level of CSPG4 expression in melanoma cells, Luo et al. found dense CpG-methylation in close proximity to the CSPG4 promotor, which corresponded with silencing of CSPG4 expression in those tumor cells [11]. Upon treatment with decitabine-dosing very similar to that used in the present study, CSPG4 upregulation was demonstrated in tumor cells that lost CSPG4 expression [11]. In accordance with the results in our study involving ovarian cancer cells, CSPG4-positivity could be induced in more than 50% of decitabine-treated melanoma cells [11]. However, the targetability of decitabine-induced CSPG4 on tumor cells has never been shown so far. The first report exploiting upregulation of target antigens for subsequent CAR-T-cell therapy was recently published. The investigators used azacitidine, a hypomethylating agent related to decitabine, to enhance the antigen-density of CD70 on malignant myeloid cells, which could then be successfully targeted by CD27-based CAR-T cells in vitro and in vivo [12]. Contrary to the present study, the target antigen CD70 was a priori uniformly expressed on the surface of leukemic blasts and not de novo induced on previously negative cells [12].

Although we observed solid decitabine-mediated CSPG4-upregulation in over 50% of ovarian cancer cells, long-lasting remissions are predicated on elimination of the whole tumor including all antigen-negative cells. Hence, the concept of targeting decitabine-upregulated CSPG4 has to be placed into a broader context encompassing strategies to simultaneously engage CSPG4-negative tumor cells. A straightforward approach would be the combinational targeting of multiple antigens. CAR-T cells targeting the common tumor antigen ErbB2 could effectively eliminate SKOV-3 cells [24]. Moreover, mesothelin represents another suitable CAR antigen on ovarian cells. CAR-T cells specific for mesothelin (MSLN) could eradicate murine xenograft models bearing SKOV-3 cells resulting in long-term remission in mice treated with MSLN-specific CAR-T cells [25]. Recently, Shu et al. published a bona fide combinational targeting approach tailored to ovarian cancer [26]. It could be demonstrated that by concomitant expression of a CAR targeting the oncofetal antigen TAG-72 (tumor-associated glycoprotein 72), which is frequently expressed in ovarian cancer, together with a CAR engaging CD47, which is overexpressed in ovarian cancer, constitutes an effective dual targeting strategy for ovarian cancer [26]. In our previous work, we developed engineering platforms to introduce a CSPG4-specific CAR together with an additional immunoreceptor into T cells via mRNA-electroporation [27]. This platform is basically expandable to generate CAR-T cells co-expressing a CSPG4-CAR together with a CAR specific for one of the antigens mentioned above to simultaneously eradicate CSPG4-negative ovarian cancer cells. Another effort to eliminate antigen-negative tumor cells relies on the recruitment of macrophages via the local CAR-activation-induced release of IL-12 from T cells redirected for universal cytokine killing (TRUCKs) [28]. Using this approach, the elimination of antigen-negative solid tumor cells could be shown in vitro and in vivo rendering CSPG4-specific TRUCKs attractive for engaging tumor cells remaining CSPG4-negative after decitabine treatment [29].

Inadvertent on-target/off-tumor toxicity constitutes a dreaded side-effect in CAR-T-cell therapy. Investigators from the National Cancer Institute (NCI) found the presence of CSPG4 mRNA at considerably lower levels in normal tissue [30]. Nevertheless, at the protein level, the expression of CSPG4 was more confined to cancer cells [31]. Immunohistochemical staining for CSPG4 of up to 30 different healthy tissues, including lung, heart, brain, skin, and the gastrointestinal tract, revealed CSPG4 expression solely in the small intestine [31]. Moreover, CSPG4-CAR-T cells co-cultured with primary epithelial cells derived from lung, kidney, and prostate cells did not exhibit cytotoxicity [32]. In addition, CSPG4 has been found to be expressed on tumor-associated pericytes, and to a significantly lower level on normal resting pericytes [33]. Against the backdrop of physiological CSPG4 expression on normal tissues, albeit at a low level, concerns may be stirred that decitabine not only increases CSPG4 expression on malignant cells but also amplifies the physiological CSPG4 expression beyond the level required for CAR activation, potentially increasing the risk for inadvertent on-target/off-tumor toxicity. Thus, comprehensive information on the impact of decitabine treatment on CSPG4 expression in healthy tissues is required. Akin to the NCI investigation mentioned above, mRNA level and protein level analyses on CSPG4 expression after decitabine treatment have to be performed before moving the concept of targeting decitabine-upregulated CSPG4, established in the present work closer to clinical translation. Using tissue specimens obtained from blood cancer patients undergoing hypomethylating therapy with decitabine, could be valuable to carry out CSPG4 expression studies at the RNA and protein levels as outlined above. These tissue specimens could be taken during active hypomethylating therapy in combination with routine biopsies. In order to generally mitigate the risk for potential on-target/off-tumor toxicity associated with targeting CSPG4, our group has developed the generation of CSPG4-specific CAR-T cells via mRNA-electroporation [17]. Owing to the evanescent expression of CARs introduced via mRNA-electroporation, toxicity originating from inadvertent on-target/off-tumor targeting will only last for a few days declining parallel to the CAR expression. Thus, CAR-T cells generated via mRNA-electroporation could mitigate the risk for inadvertent on-target/off-tumor targeting when targeting decitabine-upregulated CSPG4. In case of absent toxicity upon serial mRNA-CSPG4-CAR-T-cell infusions, the subsequent use of conventional stably transduced CAR-T cells could be conceivable.

Regarding further steps towards a potential clinical translation of targeting decitabine-upregulated CSPG4 on ovarian cancer cells using CSPG4-specific CAR-T cells, our data gathered from the SKOV-3 model provide several cues for further in vivo testing. First, optimal dosing of decitabine is the daily administration at 1 µM. Second, our time course data reveal the time points of highest CSPG4 expression to be 5 to 8 days after decitabine treatment followed by a steady decline. Third, CSPG4-CAR-T cells generated via mRNA-electroporation are capable of antigen-specific killing of decitabine-treated SKOV-3 cells. Fourth, antigen-specific killing of decitabine-treated SKOV-3 cells by CSPG4-CAR-T cells is not impaired in the presence of decitabine at 1 µM. Additionally, from our extensive experience with CAR-T cells generated via mRNA-electroporation, we know that owing to the transient nature of mRNA-electroporation, CAR expression only lasts for up to 4 (maximum 5) days [17], which perfectly matches the timeframe of highest CSPG4 expression. Coalescing all this information into a dosing regimen would require daily administration of decitabine at 1 μM for 5 days, until the simultaneous single-shot administration of CSPG4-CAR-T cells generated via mRNA-electroporation, followed by another 5 day period of decitabine application to maintain high CSPG4 expression during the time of CAR expression in RNA-CAR-T cells.

A potential limitation of the concept described here, to use decitabine-mediated upregulation on ovarian cancer cells to enable targeting by CSPG4-CAR-T cells, is posed by the tumor promoting properties of CSPG4, which might additionally foster the growth of decitabine-treated tumor cells [7]. In ovarian cancer, CSPG4 mainly promotes invasion and metastasis [20]. Moreover, CSPG4 expression may confer selected resistance to chemotherapy on ovarian cancer cells [20]. The use of CSPG4-CAR-T cells is primarily intended for metastasized patients, which generally bear a heavy tumor burden with extensive spread to the body. Limited stages of ovarian cancer are treated with surgery and chemotherapy and do not require CAR-T cells [13]. Thus, the risk of metastasis promotion by CSPG4 upregulation bears little relevance in patients harboring already multiple metastases. Moreover, CSPG4-mediated chemoresistance does even favor the use of immunotherapy, especially CAR-T cells, and no reports on CSPG4-mediated immune-evasion, which could potentially act on pathways known to impair the functionality of CAR-T-cells [34], have been published. In aggregate, the benefits of generating targetability for CSPG4-CAR-T cells via decitabine treatment clearly outweighs conceivable CSPG4-mediated tumor progress promoting actions.

Collectively, we want to shine light on the therapeutic potential of CSPG4 as an inducible secondary antigen for CAR-T-cell therapy of ovarian cancer. On account of our data, we encourage a comprehensive screening of primary ovarian cancer cells for decitabine-mediated inducibility of CSPG4. Moreover, in vivo analyses on off-tumor CSPG4 expression in response to decitabine treatment will pose an important further investigation required prior to clinical application.

## 5. Conclusions

Here, we introduced the potential of CSPG4 as an inducible target antigen for CAR-T-cell therapy of ovarian cancer cells. We show decitabine-mediated upregulation of CSPG4 on ovarian cancer cells. Additionally, we validate decitabine-upregulated CSPG4 as bona fide target antigen for CAR-T cells by demonstrating antigen-specific cytotoxicity as well as antigen-specific cytokine secretion of CSPG4-specific CAR-T cells in response to decitabine-treated SKOV-3 ovarian carcinoma cells. For the first time, we show de novo upregulation of CSPG4 in tumor cells with subsequent targeting via CSPG4-specific CAR-T cells. Building on our proof-of-concept data, targeting decitabine-mediated CSPG4 upregulation via CAR-T cells could be assayed in a variety of different tumor entities.

## Figures and Tables

**Figure 1 cancers-14-05033-f001:**
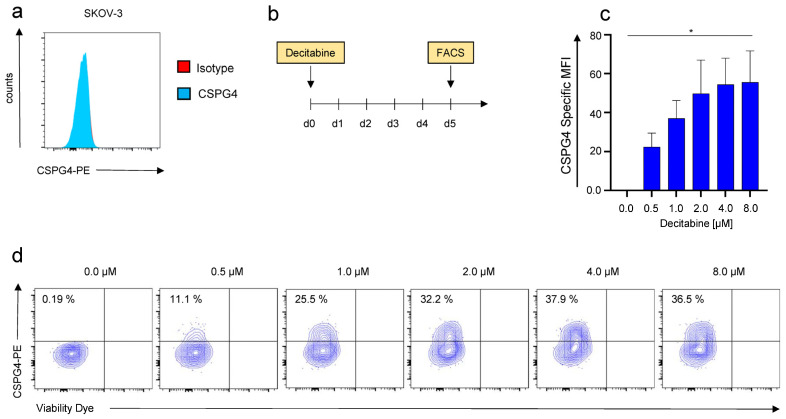
Decitabine-mediated dose-dependent upregulation of CSPG4 on SKOV-3 ovarian cancer cells. (**a**) CSPG4-staining of SKOV-3 ovarian cancer cells using a PE-labeled anti-CSPG4 antibody and a corresponding isotype control. One representative donor out of three experiments is depicted. (**b**) Schematic showing the experimental layout for the decitabine-mediated CSPG4 upregulation. (**c**,**d**) CSPG4 staining of SKOV-3 ovarian cancer cells 5 days after addition of decitabine at the indicated concentrations. (**c**) Data represent geometric means ± SEM of three runs, *p* values were calculated by Student’s *t*-test, * indicates *p* ≤ 0.05. (**d**) Contour plots of one representative run out of three independent experiments are depicted.

**Figure 2 cancers-14-05033-f002:**
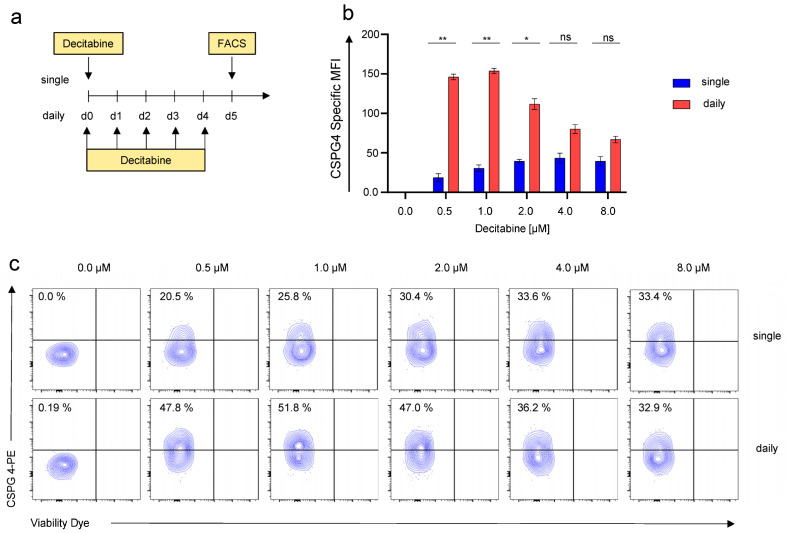
Daily application of decitabine results in greater CSPG4 induction in SKOV-3 ovarian cancer cells than single-shot application. (**a**) Schematic showing the experimental layout for the decitabine-mediated CSPG4 upregulation. SKOV-3 cells were either treated daily or once with decitabine at the indicated concentrations. For daily treatment, the culture medium was exchanged every day when adding the drug. CSPG4 upregulation was determined after 5 days. (**b**) Data represent geometric means ± SEM of three runs, *p* values were calculated by Student’s *t*-test, * indicates *p* ≤ 0.05, ** indicates *p* ≤ 0.01, and ns indicates not significant. (**c**) Contour plots of one representative run out of three independent experiments are depicted.

**Figure 3 cancers-14-05033-f003:**
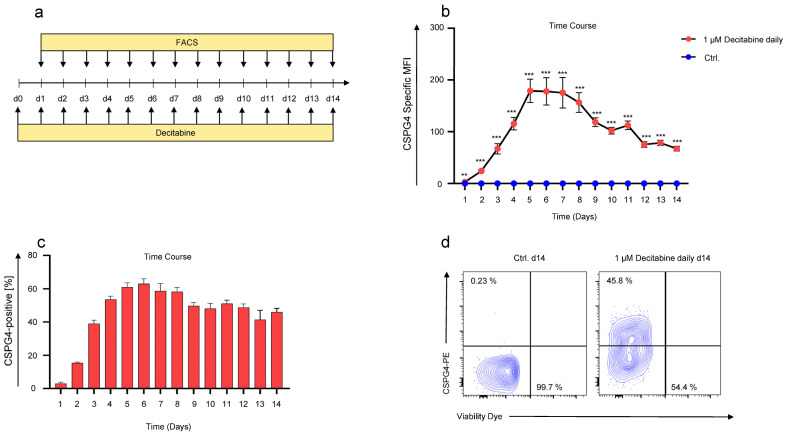
Time course of decitabine-mediated upregulation in SKOV-3 ovarian cancer cells. (**a**) Schematic showing the experimental layout for the time course experiment. SKOV-3 cells were treated daily with 1 µM of decitabine. CSPG4 staining was executed daily for 14 days. Control (Ctrl.) SKOV-3 cells were treated in the same way except for PBS addition instead of decitabine. (**b**) Data represent geometric means ± SEM of three runs, *p* values were calculated by Student’s *t*-test, ** indicates *p* ≤ 0.01 and *** indicates *p* ≤ 0.001. (**c**) Bar graph showing percentage of CSPG4-positive cells over time. Data represent geometric means ± SEM of three runs. (**d**) Contour plots showing CSPG4 staining on day 14 of SKOV-3 cells treated with PBS (Ctrl.) or 1 µM of decitabine daily. One representative experiment out of three runs.

**Figure 4 cancers-14-05033-f004:**
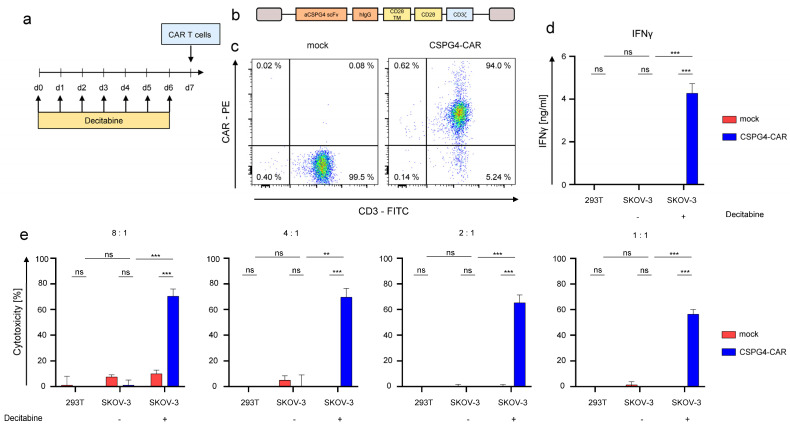
Decitabine-mediated upregulation in SKOV-3 ovarian cancer cells enables antigen-specific targeting via CSPG4-CAR-T cells. (**a**) Schematic showing the experimental layout for the experiments involving CSPG4-CAR-T cells. CSPG4-CAR-T cells were generated via RNA electroporation after 10 days of OKT3/IL-2 driven expansion. SKOV-3 cells were treated daily either with 1 µM of decitabine or PBS for 6 days prior before co-culture with CAR-T cells. (**b**) Construction plan for the second-generation CD28-co-stimulated CSPG4-specific CAR. (**c**) Expression of the CSPG4-CAR was confirmed 24 h after electroporation using a Phycoerythrin (PE)-labeled goat anti-human IgG antibody. Mock (electroporation without RNA)-T cells were used as controls. One representative donor out of three independent experiments is shown. (**d**) Interferon gamma secretion by CSPG4-CAR-T cells and mock-T cells in response to 293T cells (CSPG4-negative), SKOV-3 cells either treated with decitabine or with PBS T cells were co-cultured with target cells 24 h after electroporation at a 4:1 ratio for 48 h before supernatants were recovered and IFNγ was quantified via ELISA. Data represent means ± SEM of three donors, *p* values were calculated by Student’s *t*-test, *** indicates *p* ≤ 0.001, and ns indicates not significant. (**e**) Cytotoxicity of CSPG4-CAR-T cells and mock-T cells upon a 48 h co-culture with 293T cells, and SKOV-3 cells either treated with PBS or decitabine was assessed at the indicated effector to target ratios via an XTT-based colorimetric assay. Data represent means ± SEM of three donors, *p* values were calculated by Student’s *t*-test, *** indicates *p* ≤ 0.001, ** indicates *p* ≤ 0.01, and ns indicates not significant.

## Data Availability

Data are available upon reasonable request from the first author.

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
