# Peer review of "Decitabine-Mediated Upregulation of CSPG4 in Ovarian Carcinoma Cells Enables Targeting by CSPG4-Specific CAR-T Cells"

_cancers, 2022, doi:10.3390/cancers14205033_

Round 1
Reviewer 1 Report
The major limitation of this article is that it is an in vitro study on one ovarian cell line. No account of how these studies would translate into a real treatment is attempted. 60% positivity after 5-6 days treatment in vitro is not very encouraging. There should be some measure of the half life of this level and of the responsiveness of the still negative population of cells. What happens to any cancer stem cell component? The authors mention some of the hurdles that must be addressed before the observations reported here could be exploited as a treatment strategy. For example, off cancer target toxicity including after prolonged demethylating treatment in vivo. How might this be addressed experimentally by preclinical studies.This should be at least described if not delivered as a part of this work. The fact that the target when unregulated is likely involved in cancer progression might be a major limitation for this approach to improved CAR-T cell targeting. This should at least be discussed.
Author Response
Dear Reviewer 1,
In the present study we demonstrate up-regulation of CSPG4 on an ovarian cancer cell line upon treatment with Decitabine. We are glad that the concept of an inducible target for CAR-T cell therapy could be of interest to a broad research community. Moreover, we thank the reviewers for taking the time to review our manuscript.
Comment: The major limitation of this article is that it is an in vitro study on one ovarian cell line. No account of how these studies would translate into a real treatment is attempted.
Answer: We confirmed Decitabine-mediated CSPG4 upregulation using our established dosing regime of daily administration of 1µM decitabine for six days on Caov-3 cells and 293T cells. Caov-3 cells are a second ovarian cancer cell line, described as CSPG4-negative in the literature. Untreated Caov-3 cells exhibited negligible CSPG4 expression (supplementary Figure S1a). Upon Decitabine treatment approximately 40% of Caov3 cells showed CSPG4 expression corroborating the concept of de novo CSPG4 upregulation in ovarian cancer cells (supplementary Figure S1b). In 293T cells without Decitabine treatment no CSPG4 expression could be detected (supplementary Figure S1c), while Decitabine treatment mediated CSPG4 expression in approximately half of the treated 293T cells (supplementary Figure S1d) confirming once again the concept of Decitabine-mediated CSPG4 upregulation in neoplastic cells. We added supplemental material (supplementary Figure S1) to document those results. Furthermore, we included those results into the main text (line 244-253). Finally, we added a paragraph to the discussion section outlining a possible clinical translation of Decitabine-mediated upregulation of CSPG4, and subsequent targeting via CSPG4-specific RNA-CAR T cells (line 413-429). Here, we advocate administering Decitabine daily for ten days with a parallel single shot injection of CAR-T cells on day 5.
Comment: There should be some measure of the half life of this level and of the responsiveness of the still negative population of cells.
Answer: To analyze this question, we treated SKOV-3 cells for 14 days with 1µM Decitabine and measured CSPG4-expression daily (new Figure 3a). After two weeks, significant CSPG4 expression was still maintained with approximately half of the Decitabine treated cells showing CSPG4 expression (new Figure 3b-d). However, a tangible drop in CSPG4-expression could be observed from peak levels around days 5-8 to nearly half as much around day 12-14 (new Figure3b+c). Thus, CSPG4-expression will drop in this extended time period, but not get lost. Clinical consequences originating from the drop in CSPG4-expression are the creation of a favorable time frame (day 5 to 8 where CSPG4 expression is the highest) signifying the potentially highest therapeutic activity of RNA CSPG4-CAR T cells. Due to the transient nature of mRNA, RNA-CAR-T-cells are only active for roughly 3-4 days perfectly matching the time frame of highest CSPG4-upregulation. Thus, CAR-T cells should be administered on day 5 after Decitabine treatment where up-regulation is supposed to be highest. We exchanged Figure 3 with panels showing the extended time course until day 14, as well as the percentage of CSPG4 positive cells on day 14 as dot plots. Furthermore, we included those results into the main text (line 233-244) and highlighted the consequences of the time course for a clinical application in the discussion section (line 413-429). Mechanisms of Decitabine resistance are not subject of this work, and were not included into the Discussion section. The issue of CSPG4-negative cells after Decitabine-treatment could be potentially tackled using advanced CAR-T cell technology, such as the TRUCK technology, which has been already reported on in the discussion section (line 376-381)
Comment: What happens to any cancer stem cell component?
Answer: This study did not involve special analyses of he cancer stem cell component, but focuses on highlighting a general concept of CSPG4 as an inducible target for CAR-T cell therapy of ovarian cancer.
Comment: The authors mention some of the hurdles that must be addressed before the observations reported here could be exploited as a treatment strategy. For example, off cancer target toxicity including after prolonged demethylating treatment in vivo. How might this be addressed experimentally by preclinical studies. This should be at least described if not delivered as a part of this work.
Answer: This is a very important point as it directly affects patients´ safety. We added a paragraph to the discussion section (line 400-404) outlining a possible screening for off-tumor CSPG4 up-regulation based on histopathological specimens obtained from patients with blood cancer undergoing hypomethlyting therapy.
Comment: The fact that the target when unregulated is likely involved in cancer progression might be a major limitation for this approach to improved CAR-T cell targeting. This should at least be discussed.
Answer: To acknowledge this potentially important limitation, we added a paragraph to the Discussion section (line 430-446). Importantly, CSPG4 promotes tumor progression by facilitating metastasis formation and invasion. CSPG4-CAR-T cells are intended for the use in patients with systemic disease and present metastases. So far, CSPG4 has not been implicated in immune evasion, which could impair CSPG4-directed CAR-T cell therapy. We clearly see the benefits of generating targetability for CSPG4-CAR T cells via Decitabine treatment outweighing potential CSPG4-mediated tumor promoting actions.

Reviewer 2 Report
Dennis Christoph Harrer and colleagues present a high quality and well-written experimental manuscript that describes decitabine-mediated upregulation of CSPG4 in ovarian carcinoma cells that enables targeting by CSPG4-specific CAR-T cells.
Authors aimed to evaluate the potential of the well-established melanoma antigen chondroitin sulfate proteoglycan 4 (CSPG4) as an inducible antigen in ovarian cancer cells, using CSPG4-negative SKOV-3 ovarian cancer cells as a model. Based on the hypomethylating activity of the FDA-approved drug Decitabine, they refined a protocol to upregulate CSPG4 in the majority of Decitabine-treated SKOV-3 cells. CSPG4-specific CAR-T cells generated by mRNA-electroporation showed CSPG4-directed cytokine secretion and cytotoxicity towards Decitabine-treated SKOV-3.
Authors claim that they generated proof-of-concept data paving the way for the further exploration of CSPG4 as an inducible antigen for CAR-T cells in ovarian cancer. They also raised awareness for CSPG4 as a potentially inducible antigen in solid tumors.
Authors’ findings support the notion of the therapeutic potential of CSPG4 as an inducible secondary antigen for CAR-T-cell therapy of ovarian cancer. They encourage a comprehensive screening of primary ovarian cancer cells for Decitabine-mediated inducibility of CSPG4. Moreover, in-vivo analyses on off-tumor CSPG4 expression in response to Decitabine treatment will pose an important further investigation required prior to clinical application.
Finally, authors conclude that they introduced the potential of CSPG4 as an inducible target antigen for CAR-T-cell therapy of ovarian cancer cells. They showed Decitabine-mediated upregulation of CSPG4 on ovarian cancer cells. Additionally, they validated Decitabine-upregulated CSPG4 as bona fide target antigen for CAR-T cells by demonstrating antigen-specific cytotoxicity as well as antigen-specific cytokine secretion of CSPG4-specific CAR-T cells in response to Decitabine-treated SKOV-3 ovarian carcinoma cells. For the first time, they showed de-novo upregulation of CSPG4 in tumor cells with subsequent targeting via CSPG4-specific CAR-T cells. Building on these proof-of-concept data, targeting Decitabine-mediated CSPG4-upregulation via CAR-T cells could be assayed in a variety of different tumor entities.
Overall, the manuscript is highly valuable for the scientific community and should be accepted for publication.
======================
Other comments to authors:
1) Please check for typos throughout the manuscript.
2) Line 31. “aimed at” -> “aimed to”
3) Authors are kindly encouraged to cite the following article that describes various aspects of CAR-T cell functioning. DOI: 10.3390/cancers14041078
Author Response
Dear Reviewer 2,
In the present study we demonstrate up-regulation of CSPG4 on an ovarian cancer cell line upon treatment with Decitabine. We are glad that the concept of an inducible target for CAR-T cell therapy could be of interest to a broad research community. Moreover, we thank the reviewers for taking the time to review our manuscript.
1) Please check for typos throughout the manuscript.
Answer: We scrutinized the manuscript and corrected remaining typos.
2) Line 31. “aimed at” -> “aimed to”
Answer: We corrected the typo as suggested.
3) Authors are kindly encouraged to cite the following article that describes various aspects of CAR-T cell functioning. DOI: 10.3390/cancers14041078
Answer: We included this reference in the discussion section.
